# Hypoxemia, hypoglycemia and IMCI danger signs in pediatric outpatients in Malawi

André Thunberg[1,2]*, Beatiwel Zadutsa[3], Everlisto Phiri[3], Carina King[1,4], Josephine Langton[5], Lumbani Banda[3], Charles Makwenda[3], Helena Hildenwall[1,2,6]

1 Department of Global Public Health, Karolinska Institutet, Stockholm, Sweden, 2 Astrid Lindgren Children's Hospital, Karolinska University Hospital, Stockholm Sweden, 3 Parent and Child Health Initiative, Lilongwe, Malawi, 4 Institute for Global Health, University College London, London, England, 5 Department of Paediatrics, College of Medicine, Blantyre, Malawi, 6 Department of Clinical Science, Intervention and Technology, Karolinska Institutet, Stockholm, Sweden

* andre.thunberg@ki.se

**Data Availability Statement:** Unidentified data has been uploaded to the public repository Dataverse and can be accessed here: https://doi.org/10.7910/DVN/0YWX8J.

## Abstract

Hypoxemia and hypoglycemia are known risks for mortality in children in low-income settings. Routine screening with pulse oximetry and blood glucose assessments for outpatients could assist in early identification of high-risk children. We assessed the prevalence of hypoglycemia and hypoxemia, and the overlap with Integrated Management of Childhood Illness (IMCI) general danger signs, among children seeking outpatient care in Malawi. A cross-sectional study was conducted at 14 government primary care facilities, four rural hospitals and one district referral hospital in Mchinji district, Malawi from August 2019—April 2020. All children aged 0–12 years seeking care with an acute illness were assessed on one day per month in each facility. Study research assistants measured oxygen saturation using Lifebox LB-01 pulse oximeter and blood glucose was assessed with AccuCheck Aviva glucometers. World Health Organization definitions were used for severe hypoglycemia (<2.5mmol/l) and hypoxemia (SpO$_2$ <90%). Moderate hypoglycemia (2.5–4.0mmol/l) and hypoxemia (SpO$_2$ 90–93%) were also calculated and prevalence levels compared between those with and without IMCI danger signs using chi2 tests. In total 2,943 children were enrolled, with a median age of 41 (range: 0–144) months. The prevalence of severe hypoxemia was 0.6% and moderate hypoxemia 5.4%. Severe hypoglycemia was present in 0.1% of children and moderate hypoglycemia in 11.1%. IMCI general danger signs were present in 29.3% of children. All severely hypoglycemic children presented with an IMCI danger sign (p <0.001), but only 23.5% of the severely hypoxemic and 31.7% of the moderately hypoxemic children. We conclude that while the prevalence of severe hypoxemia and hypoglycemia were low, moderate levels were not uncommon and could potentially be useful as an objective tool to determine referral needs. IMCI danger signs identified hypoglycemic children, but results highlight the challenge to detect hypoxemia. Future studies should explore case management strategies for moderate hypoxemia and hypoglycemia.

**Funding:** The study was funded by grants from the Vetenskapsrådet SE (2017-05579) HH, Laerdal Foundation (40348) HH and the Einhorn Family Foundation HH. The funders had no role in study design, data collection and analysis, decision to publish or preparation of the manuscript.

**Competing interests:** We declare no financial, personal, or professional competing interests that have influenced the work.

## Introduction

Global child mortality has more than halved since 1990, but still 5.2 million children died before the age of five in 2019, with treatable infectious diseases remaining the leading post-neonatal cause of death [1]. The World Health Organization's (WHO) Integrated Management of Childhood Illness (IMCI) was first introduced in the mid-1990's and has since been implemented to some extent in over 100 countries [2]. This program aims to reduce child mortality through the timely identification of common infections, and increased access to anti-biotics, antimalarial treatment and oral rehydration solution and zinc [3, 4]. Full implementation of IMCI has been linked with the achievement of the Millennium Development Goal 4 which aimed to reduce the under-five mortality rate by two thirds between 1990 and 2015 [5].

However, IMCI does not include any components of emergency case management, and primary healthcare facilities in low-income countries commonly lack the resources to treat severely ill children and instead refer these cases to hospitals [6]. Substantial challenges in the identification of severely ill children have been reported under IMCI, potentially leading to missed referrals [7]. In addition, completion of referrals are commonly complicated by family circumstances, costs and transportation issues [8], as well as gender norms [9], leading to delayed or incomplete referrals [10].

Early objective identification of children at higher risk of mortality could improve referral practices, both through emphasizing to caregivers the importance of attending the hospital, and by ensuring that children reach higher level facilities before they are critically ill. Hypoxemia and hypoglycemia are both known to be associated with poor outcomes [11, 12] but may go clinically undetected [13]. A study in Malawi reported that more than half of the hypoxemic pneumonia cases in children would not have been referred from primary care, using only the IMCI guidelines in the absence of a pulse oximeter [14]. Furthermore, the definitions of hypoxemia and hypoglycemia are topics of debate with studies demonstrating increased mortality also among children with moderately low saturation and blood glucose values [11, 15–19].

More frequent use of pulse oximetry and blood glucose assessments within higher mortality outpatient settings could improve outcomes through earlier identification and referral of those with moderately low oxygen saturation values and blood glucose concentrations. To inform referral strategies, the burden of severe illness needs to be determined alongside the utility of the current IMCI danger signs to identify children at risk. Therefore, we aimed to assess the prevalence of severe and moderate hypoxemia and hypoglycemia in children presenting to outpatient settings in Mchinji district, Malawi, and to explore the overlap between different levels of hypoxemia and hypoglycemia and IMCI danger signs.

## Materials and methods

We conducted a cross sectional study of children aged 0 to 12 years presenting to outpatient facilities with an acute illness in Mchinji district, Malawi, from August 2019 to April 2020. The study was nested within a district wide cohort study (**E**mergency paediatric treatment and **re**ferral in **M**alawi **i**n frontline healthcare **s**ettings–EREMISS study), which assessed the outcomes of children being referred from primary care health facilities. While the EREMISS study only enrolled children who were referred from primary care facilities to higher level care, the current study was run in parallel to provide data on the overall prevalence of hypoxemia and hypoglycemia among children seeking outpatient care in the district.

### Setting

Mchinji district is located in Malawi's central region and with an under-five population of approximately 90,000 in 2018 [20]. The under-five mortality rate was 123/1,000 livebirths in

the 2015–16 Demographic Health Survey [21]. Data was collected at the 19 public health facilities, consisting of: 14 government health centers, four rural hospitals run by the Christian Health Association of Malawi (CHAM) and the district hospital. Health centers provide no pediatric inpatient care while hospitals both provide outpatient services and inpatient care. Patients were only enrolled from the outpatient department of the included hospitals. All care for children is provided free of charge at government facilities. Malawi implemented IMCI guidelines in 2000 [22].

## Participant recruitment

Data was collected for one full working day per month at each of the study facilities. Study research assistants were present to collect data during 'normal' facility outpatient clinic times, usually between 8am– 3pm. Data collection days were randomly determined, using a simple random number generator with replacement in Microsoft Excel. This was done to ensure that any potential differences in patient load and/or type of diagnosis depending on the day of the week would be spread randomly across study facilities. Guardians of all children from 0 up to 12 years of age, inclusive, who sought care due to an acute illness (i.e., excluding children coming for immunization or growth control) were approached for recruitment.

## Data collection

Children and their guardians were informed about the study when they were in the facility waiting area. Before seeing the clinician, guardians were approached by a study research assistant to ask for their consent, both to participate in the study and to have a blood glucose test done. Once consent was given, the research assistant assessed the child's oxygen saturation using a Lifebox LB-01 pulse oximeter (Acare Technology Co. Ltd), with pediatric and universal clip probes. Research assistants were trained to use the child's big toe for measurements and to wait for a stable waveform before collecting the saturation value. Blood glucose concentrations were measured after the oxygen saturation had been assessed. A capillary blood sample was collected after pricking the child's finger and the glucose concentration was analyzed using a AccuCheck Aviva (Roche Diabetes Care, Inc) point of care device reporting results in mmol/l. The results were written in the child's health passport, and clinical staff were immediately alerted if the blood glucose concentration was <2.5 mmol/l or if the oxygen saturation was <90%.

After these measurements, the child was assessed by routine clinic staff. Data on IMCI general danger signs i.e. being unable to eat/drink, vomits everything, unconscious, sleepy, lethargic and convulsions [3] were then extracted by the research assistant from the child's health passport. While the IMCI danger signs are created for children below five years of age, there is a lack of validated assessments tools and routine guidelines for children aged 5–12 years, and therefore health workers in this setting commonly use IMCI for this age group. We therefor chose to extract the same clinical information for all children. Other variables collected were age and sex, and from the health passport any documented nutritional status, respiratory rate, temperature, presence of chest indrawing and the health worker diagnosis. The additional variables were chosen since commonly assessed by health workers and relevant for the potential relation with hypoxemia.

Data was collected electronically using Open Data Kit (ODK) on Android tablets and uploaded daily to an ODK database. Data queries were raised and resolved with regular communication between the Monitoring and Evaluation Officers, investigators and data manager. All patients were managed routinely by the facility staff according to needs and facility standards.

Prior to the study start, all participating healthcare providers underwent re-training in IMCI to ensure that minimum standard of care was being provided. A total of 20 research assistants were recruited locally and assigned different facilities within the study area. Research assistants had no formal clinical training but underwent one week training including: study procedures, conducting pulse oximetry and blood glucose testing and filling case report forms accurately. Following training, the study procedures were piloted for one week; continual mentorship and supervision was provided by the study Monitoring and Evaluation Officers and project manager in close communication with the study investigators, to correct any issues in protocol implementation.

## Data management and analysis

Oxygen saturation was categorized into normal ($SpO_2 > = 94\%$), moderate hypoxemia ($SpO_2$ 90–93%) and severe hypoxemia ($SpO_2 < 90\%$). Blood glucose concentration were classified as normal (4.1–11.0 mmol/l), moderate hypoglycemia (2.5–4.0 mmol/l) and severe hypoglycemia (<2.5mmol/l). Since nutritional assessment was not recorded for all children, we used the <2.5mmol/l as the primary hypoglycemia definition for the whole group but the prevalence of blood glucose concentrations <3 mmol/l was also calculated for those assessed to be malnourished. During data checks we noted that two research assistants reported unreasonably high proportions of hypoxemia, suggestive of measurement quality issues. We decided to exclude their $SpO_2$ results (n = 306) from the primary analysis but conducted a sensitivity analysis which retained them (presented in S1 Table).

Patient characteristics were described by facility type, oxygen saturation and blood glucose level using proportions. Malnutrition was defined using weight-for-age Z-score (WAZ; moderately malnourished -3 to -2 SD; severely malnourished <-3 SD using WHO growth charts for children <5 years old and UK growth charts for children aged 5–12), or health worker's clinical diagnosis. Differences in clinical characteristics between those with and without danger signs were compared using chi2 and Fishers exact tests.

As a quality control of the reported danger signs, we conducted a sensitivity analysis comparing the prevalence of danger signs between children with a positive malaria Rapid Diagnostic Test (danger signs expected) and children diagnosed with scabies only (danger signs not expected). Smoothed graphs were plotted to show the seasonal variation in clinical presentation, using a 2-week weighted moving averages. All data processing and analysis was done using Stata 12.

## Ethical statement

The study was approved by the Malawi College of Medicine Research and Ethics Committee (reference: P11/18/25389). Guardians were informed about the study while in the waiting area and provided verbal consent for their minors to participate in the study prior to any data was collected. Due to literacy levels, study information was given verbally in Chichewa, and consent given verbally–the informed consent was subsequently recorded in the electronic data collection form. The consent covered both the collection of data from patient health passports and the assessment of blood glucose concentrations and oxygen saturation. Refusal to participate had no impact on the care provided to the patients and information sheets with study contact information were available for guardians to take with them.

## Results

A total of 2,943 children aged 0–12 years were enrolled as they presented to health centers (2,343/2,943, 79.6%) and hospitals (601/2,943, 20.4%)–Table 1. There was a similar proportion

**Table 1. Baseline characteristics of recruited children.**

| | Hospital | Health center | Total |
|---|---|---|---|
| | N = 601 | N = 2,342 | N = 2,943 |
| | n (%) | n (%) | n (%) |
| **Age** | | | |
| <2 months | 8 (1.3) | 27 (1.2) | 35 (1.2) |
| 2–11 months | 102 (17.0) | 371 (15.8) | 473 (16.1) |
| 12–59 months | 320 (53.2) | 999 (42.7) | 1,319 (44.8) |
| 5–12 years | 171 (28.5) | 945 (40.4) | 1,116 (37.9) |
| **Sex** | | | |
| Male | 292 (48.6) | 1,136 (48.5) | 1,428 (48.5) |
| Female | 309 (51.4) | 1,206 (51.5) | 1,515 (51.5) |
| **Socio-economic factors** | | | |
| **-  Education of mother** | | | |
| No formal education | 14 (2.3) | 185 (7.9) | 199 (6.8) |
| Primary | 357 (59.4) | 1,752 (74.8) | 2,109 (71.7) |
| Secondary/further | 226 (37.6) | 400 (17.1) | 626 (21.3) |
| Missing | 4 (0.7) | 5 (0.2) | 9 (0.3) |
| **-  Mother's marital status** | | | |
| Married | 541 (90.0) | 2,076 (88.6) | 2,617 (88.9) |
| Never married | 22 (3.7) | 41 (1.8) | 63 (2.1) |
| Divorced/separated | 28 (4.7) | 206 (8.8) | 234 (8.0) |
| Widowed | 10 (1.7) | 17 (0.7) | 27 (0.9) |
| Missing | 0 (0) | 2 (0.1) | 2 (0.1) |
| **Nutritional status (WAZ and clinical)** | | | |
| Well nourished | 524 (87.2) | 1,147 (49.0) | 1.671 (56.8) |
| Moderately malnourished | 37 (6.2) | 117 (5.0) | 154 (5.2) |
| Severely malnourished | 24 (4.0) | 145 (6.2) | 169 (5.7) |
| Missing | 16 (2.7) | 933 (40.0) | 949 (32.3) |
| **Diagnoses** [*] | | | |
| Non-infectious | 85 (14.1) | 312 (13.3) | 397 (13.5) |
| **-  Infections** | | | |
| Gastroenteritis | 36 (6.0) | 152 (6.5) | 188 (6.4) |
| Malaria | 315 (52.4) | 1,132 (48.3) | 1,447 (49.2) |
| Non-pneumonia respiratory tract infection | 117 (19.5) | 595 (25.4) | 712 (24.2) |
| Pneumonia | 9 (1.5) | 23 (1.0) | 32 (1.1) |
| Sepsis | 60 (10.0) | 185 (7.9) | 245 (8.3) |
| Other infection | 33 (5.5) | 325 (13.9) | 358 (12.2) |

[*] Children may have more than one diagnosis and therefore the total number of diagnoses exceeds the total number of children.

of boys and girls (48.5% and 51.5%, respectively), and this was comparable at both hospital and health centers. The most common diagnosis was malaria (49.2%), followed by respiratory tract infections (24.2%).

## Blood glucose assessment

Blood glucose was recorded in 99.1% of children (2,917/2,943; 1 child was too agitated, 3 care-givers refused assessment and in 22 cases there was a lack of test strips). The median blood glucose concentration was 5.3 mmol/l (IQR 4.5–6.0). Table 2 presents the characteristics of

**Table 2. Prevalence of different blood glucose concentrations by background and clinical characteristics.**

| | Severe Hypoglycemia <2.5 mmol/l | Moderate hypoglycemia 2.5–4.0 mmol/l | Normoglycemia 4.1–11.0 mmol/l | Hyperglycemia >11 mmol/l | Total |
|---|---|---|---|---|---|
| | N = 4 | N = 323 | N = 2,579 | N = 11 | N = 2,917 |
| | n (%) | n (%) | n (%) | n (%) | n (%) |
| **Age** | | | | | |
| <2 month | 1 (25.0) | 3 (0.9) | 29 (1.1) | 1 (9.1) | 34 (1.2) |
| 2–11 months | 0 (0.0) | 29 (9.0) | 439 (17.0) | 0 (0.0) | 468 (16.0) |
| 12–59 months | 2 (50.0) | 143 (44.3) | 1,150 (44.6) | 9 (81.8) | 1,304 (44.7) |
| 5–12 years | 1 (25.0) | 148 (45.8) | 961 (37.3) | 1 (9.1) | 1,111 (38.1) |
| **Sex** | | | | | |
| Male | 2 (50.0) | 140 (43.3) | 1,276 (49.5) | 3 (27.3) | 1,421 (48.7) |
| Female | 2 (50.0) | 183 (56.7) | 1,303 (50.5) | 8 (72.7) | 1,496 (51.3) |
| **Facility type** | | | | | |
| Hospital | 0 (0.0) | 44 (13.6) | 555 (21.5) | 2 (18.2) | 601 (20.6) |
| Health center | 4 (100.0) | 279 (86.4) | 2,024 (78.5) | 9 (81.8) | 2,316 (79.4) |
| **Nutritional status (WAZ and clinical)** | | | | | |
| Well nourished | 2 (50.0) | 179 (55.4) | 1,481 (57.4) | 7 (63.6) | 1,669 (57.2) |
| Moderate malnourished | 0 (0.0) | 24 (7.4) | 130 (5.0) | 0 (0.0) | 154 (5.3) |
| Severely malnourished | 0 (0.0) | 22 (6.8) | 144 (5.6) | 3 (27.3) | 169 (5.7) |
| Missing | 2 (50.0) | 98 (30.3) | 824 (32.0) | 1 (9.1) | 925 (31.7) |
| **Any IMCI danger sign*** | 4 (100.0) | 112 (34.7) | 725 (28.1) | 5 (45.5) | 846 (29.0) |
| Unable to drink/feed | 2 (50.0) | 59 (18.3) | 317 (12.3) | 0 (0.0) | 378 (13.0) |
| Vomits everything | 1 (25.0) | 39 (12.1) | 381 (14.8) | 1 (9.1) | 422 (14.5) |
| Convulsions | 0 (0.0) | 1 (0.3) | 16 (0.6) | 0 (0.0) | 17 (0.6) |
| Unconscious | 0 (0.0) | 5 (1.6) | 36 (1.4) | 0 (0.0) | 41 (1.4) |
| Sleepy/lethargic | 2 (50.0) | 61 (18.9) | 331 (12.8) | 4 (36.4) | 398 (13.6) |
| **Clinical signs/test** | | | | | |
| Chest indrawing | 0 (0.0) | 2 (0.7) | 32 (1.2) | 1 (9.1) | 35 (1.2) |
| Temperature = >37.5°C ** | 1 (25.0) | 98 (30.3) | 784 (30.4) | 7 (63.6) | 890 (30.5) |
| Malaria RDT positive | 3 (75.0) | 175 (54.2) | 1,245 (48.3) | 6 (54.6) | 1,429 (49.0) |
| **Diagnoses by clinician ***** | | | | | |
| Non-infectious | 0 (0.0) | 35 (10.8) | 356 (13.8) | 1 (9.1) | 392 (13.4) |
| - Infections | | | | | |
| Gastroenteritis | 0 (0.0) | 25 (7.7) | 162 (6.3) | 0 (0.0) | 187 (6.4) |
| Malaria | 3 (75.0) | 166 (51.4) | 1,260 (48.8) | 4 (36.4) | 1,433 (49.1) |
| Non-pneumonia respiratory tract infection | 0 (0.0) | 86 (26.6) | 619 (24.0) | 4 (36.4) | 709 (24.3) |
| Pneumonia | 0 (0.0) | 3 (0.9) | 27 (1.1) | 0 (0.0) | 30 (1.0) |
| Sepsis | 0 (0.0) | 19 (5.9) | 223 (8.7) | 1 (9.1) | 243 (8.3) |

(*Continued*)

**Table 2.** (Continued)

| | Severe Hypoglycemia <2.5 mmol/l | Moderate hypoglycemia 2.5–4.0 mmol/l | Normoglycemia 4.1–11.0 mmol/l | Hyperglycemia >11 mmol/l | Total |
|---|---|---|---|---|---|
| | N = 4 | N = 323 | N = 2,579 | N = 11 | N = 2,917 |
| | n (%) | n (%) | n (%) | n (%) | n (%) |
| Other infections**** | 1 (25.0) | 64 (19.8) | 290 (11.2) | 3 (27.3) | 358 (12.2) |

* The same child could have more than one danger sign.

** 2,298 children had a temperature taken. 2,295 had temperature and glucose result.

*** The same child could have more than one diagnosis.

**** 1 child with hypoglycemia was diagnosed with fever of unknown cause and included in "other infections".

patients, depending on the level of the blood glucose concentration. The prevalence of severe hypoglycemia (<2.5 mmol/l) was 0.1% (95% CI 0.03–0.27) and moderate hypoglycemia (2.5–4.0 mmol/l) 11.1% (95% CI 9.9–12.2). All the children with severe hypoglycemia and 34.7% with moderate hypoglycemia (n = 112) presented with at least one IMCI danger sign with lethargy and inability to drink being more common among the severely/moderately hypoglycemic than in the normoglycemic group (p<0.01 for both). The most common diagnosis for both severely and moderately hypoglycemic children was malaria (75% and 51.4%, respectively). Of the children classified as severely malnourished, 1.8% (3/169) had a blood glucose concentration <3.0mmol/l and 11.2% (19/169) had a blood glucose of 3.0–4.0mmol/l.

## Oxygen saturation measurement

Table 3 demonstrates the patient characteristics depending on their oxygen saturation. Three patients lacked SpO$_2$-results due to caregiver refusal, and 306 suspected error measurements were excluded, giving a total of 2,634 (89.5%) children with a SpO$_2$-result. Of these, a stable SpO$_2$ curve was not achieved in 4.7% (n = 124) cases, and these are reported separately. An unstable curve was more common in children <12 months compared to older children (8.4% vs 4.0%, p<0.01).

The median SpO$_2$ was 97% (IQR 96–98%). The prevalence of severe hypoxemia (SpO$_2$ <90%) was 0.6% (95% CI 0.3–0.9) and moderate hypoxemia was present in 5.4% (95% CI 4.6–6.3) of children. None of the children with hypoxemia had recorded chest indrawing, and only one had been assessed with tachypnea. Malaria was the most common diagnosis among severely hypoxemic children (58.8%), while no child with hypoxemia had been diagnosed by the routine healthcare worker as having pneumonia. The sensitivity analysis with the excluded SpO$_2$ results did not show any difference in the presence of danger sign among the severely and moderately hypoxemic patients (S1 Table).

## General danger signs

Table 4 presents the clinical characteristics of the children, according to the presence of IMCI danger signs. There was no significant difference between children aged <5 years and children aged 5–12 years in terms of presence of any IMCI danger sign (p = 0.072). Of all children presenting with a danger sign, 512 children had only one danger sign and 336 had two or more. At least one of the IMCI danger signs was present in 28.8% (95%CI 27.2–30.5) of children. If including chest indrawing as an indicator for referral, the proportion increased to 29.3% (n = 861), with chest indrawing documented in only 35/2,943 children. The prevalence of high temperature and chest indrawing was more common in the group with danger signs than in those without any danger sign (p<0.001 for both).

**Table 3. Prevalence of oxygen saturation levels depending on background and clinical characteristics.**

|  | Unstable curve | Severe Hypoxemia <90% | Moderate hypoxemia 90–93% | Normal 94–100% | Total |
|---|---|---|---|---|---|
|  | N = 124 | N = 17 | N = 142 | N = 2,351 | N = 2,634 |
|  | n (%) | n (%) | n (%) | n (%) | n (%) |
| **Age** |  |  |  |  |  |
| <2month | 4 (3.2) | 0 (0.0) | 7 (4.9) | 16 (0.7) | 27 (1.0) |
| 2–11 months | 32 (25.8) | 1 (5.9) | 41 (28.9) | 334 (14.2) | 408 (15.5) |
| 12–59 months | 71 (57.3) | 10 (58.8) | 70 (49.3) | 1,029 (43.8) | 1,180 (44.8) |
| 5–12 years | 17 (13.7) | 6 (35.3) | 24 (16.9) | 972 (41.3) | 1,019 (38.7) |
| **Sex** |  |  |  |  |  |
| Male | 64 (51.6) | 10 (58.8) | 75 (52.8) | 1,138 (48.4) | 1,287 (48.9) |
| Female | 60 (48.4) | 7 (41.2) | 67 (47.2) | 1,213 (51.6) | 1,347 (51.1) |
| **Facility type** |  |  |  |  |  |
| Hospital | 10 (8.1) | 4 (23.5) | 65 (45.8) | 522 (22.2) | 601 (22.8) |
| Health center | 114 (91.9) | 13 (76.5) | 77 (54.2) | 1,829 (77.8) | 2,033 (77.2) |
| **Nutritional status (WAZ and clinical)** |  |  |  |  |  |
| Well nourished | 38 (30.7) | 11 (64.7) | 107 (75.4) | 1,316 (56.0) | 1,472 (55.9) |
| Moderate malnourished | 4 (3.2) | 2 (11.8) | 6 (4.2) | 118 (5.0) | 130 (4.9) |
| Severely malnourished | 1 (0.8) | 0 (0.0) | 7 (4.9) | 132 (5.6) | 140 (5.3) |
| Missing | 81 (65.3) | 4 (23.5) | 22 (15.5) | 785 (33.4) | 892 (34.9) |
| **Any IMCI danger sign** | 20 (16.1) | 4 (23.5) | 45 (31.7) | 685 (29.1) | 754 (28.6) |
| Unable to drink/breastfeed | 10 (8.1) | 1 (5.9) | 23 (16.2) | 272 (11.6) | 306 (11.6) |
| Vomits everything | 14 (11.3) | 2 (11.8) | 21 (14.8) | 340 (14.5) | 377 (14.3) |
| Convulsions | 0 (0.0) | 0 (0.0) | 3 (2.1) | 14 (0.6) | 17 (0.7) |
| Unconscious | 3 (2.4) | 1 (5.9) | 3 (2.1) | 22 (0.9) | 29 (1.1) |
| Sleepy/lethargic | 10 (8.1) | 1 (5.9) | 26 (18.3) | 354 (15.1) | 391 (14.8) |
| **Clinical signs/test** |  |  |  |  |  |
| Chest indrawing | 2 (1.6) | 0 (0.0) | 5 (3.5) | 22 (0.9) | 29 (1.1) |
| Tachypnea for age (IMCI)* | 1 (0.8) | 1 (5.9) | 1 (0.7) | 47 (2.0) | 50 (1.9) |
| Temperature = >37.5˚C ** | 30 (24.2) | 5 (29.4) | 65 (45.8) | 718 (30.5) | 818 (31.1) |
| Malaria RDT positive | 46 (37.1) | 10 (58.8) | 62 (43.7) | 1,218 (51.8) | 1,336 (50.7) |
| **Diagnoses by clinician***** |  |  |  |  |  |
| Non-infectious | 13 (10.5) | 2 (11.8) | 16 (11.3) | 290 (12.3) | 321 (12.2) |
| **- Infections** |  |  |  |  |  |
| Gastroenteritis | 10 (8.1) | 3 (17.7) | 9 (6.3) | 133 (5.7) | 155 (5.9) |
| Malaria | 47 (37.9) | 10 (58.8) | 61 (43.0) | 1,240 (52.7) | 1,358 (51.6) |
| Non-pneumonia respiratory tract infection | 48 (38.7) | 4 (23.5) | 40 (28.2) | 549 (23.4) | 641 (24.3) |
| Pneumonia | 3 (2.4) | 0 (0.0) | 5 (3.5) | 24 (1.0) | 32 (1.2) |
| Sepsis | 11 (8.9) | 0 (0.0) | 16 (11.3) | 214 (9.1) | 241 (9.2) |
| Other infections | 11 (8.9) | 2 (11.8) | 9 (6.3) | 248 (10.6) | 270 (10.3) |

* Tachypnea defined as RR >60 for children <2 month, >50 2–12 months, >40 12–60 months, <30 >60 months according to IMCI guidelines. RR assessed in 263 out of the 2,634 children included in SpO$_2$ measurement.

** Temperature was measured in 2,024 out of the 2634 tested for blood glucose.

*** The same child could have more than one diagnosis.

The group of children presenting with at least one danger sign included all severely hypoglycemic children and 34.7% with moderate hypoglycemia. Only 23.5% of children with severe hypoxemia and 31.7% of children with moderate hypoxemia had a danger sign. Of all children

**Table 4. Clinical characteristics by presence of any IMCI danger sign.**

| | IMCI general danger sign present | IMCI general danger sign not present | P-value |
|---|---|---|---|
| | N = 848 | N = 2,095 | |
| | n (%) | n (%) | |
| **Age** | | | |
| <2month | 4 (0.5) | 31 (1.5) | **p<0.001** |
| 2–11 months | 93 (11.0) | 380 (18.1) | |
| 12–59 months | 408 (48.1) | 911 (43.5) | |
| 5–12 years | 343 (40.5) | 773 (36.9) | |
| **Temperature** * | | | |
| Low <36.5˚C | 180 (21.2) | 460 (22.0) | p<0.001 |
| Normal 36.5˚C– 37.4˚C | 215 (25.4) | 551 (26.3) | |
| High = >37.5˚C | 360 (42.5) | 532 (25.4) | |
| Missing | 93 (11.0) | 552 (26.4) | |
| **Respiratory signs** | | | |
| Tachypnea for age** | 38 (4.5) | 63 (3.0) | p = 0.088 |
| RR not done/error | 718 (84.7) | 1,826 (87.2) | |
| Chest indrawing | 22 (2.6) | 13 (0.6) | p <0.001 |
| **Oxygen saturation (%)** | | | |
| Normal $SpO_2$ (94–100) | 685 (80.8) | 1,666 (79.5) | p = 0.018 |
| Moderate hypoxemia (90–93) | 45 (5.3) | 97 (4.6) | |
| Severe Hypoxemia (<90) | 4 (0.5) | 13 (0.6) | |
| Unable to obtain stable $SpO_2$ wave | 20 (2.4) | 104 (5.0) | |
| Excluded $SpO_2$-results*** | 94 (11.1) | 215 (10.3) | |
| **Blood glucose concentration (mmol/l)** | | | |
| Severe Hypoglycemia (<2.5) | 4 (0.5) | 0 (0.0) | p<0.001 |
| Moderate hypoglycemia (2.5–4.0) | 112 (13.2) | 211 (10.1) | |
| Normal b-glucose (4.1–11) | 725 (85.5) | 1,854 (88.5) | |
| Hyperglycemia (>11) | 5 (0.6) | 6 (0.3) | |
| No blood glucose result**** | 2 (0.2) | 24 (1.2) | |
| **Nutritional status** | | | |
| Well nourished | 459 (54.1) | 1,212 (57.9) | p<0.001 |
| Moderately malnourished | 60 (7.1) | 94 (4.5) | |
| Severely malnourished | 70 (8.3) | 99 (4.7) | |
| Missing | 259 (30.1) | 690 (32.9) | |

*Temperature was assessed in 2,298.

** Respiratory rate was assessed in 399 children.

*** 306 excluded (every assessment from two research assistants with frequent unexpected results). 3 missing due to refusal by caregiver.

**** No result due to refusal by caregiver (3), lack of test strips (22) and 1 child too agitated.

with severe or moderate hypoxemia (n = 159), there was no significant difference in the presence of a danger sign between children below 5 years of age (29.5%) and children aged 5–12 years (36.7%, p = 0.441). Similarly, there was no significant difference between age groups in the presence of danger signs among children with severe or moderate hypoglycemia (n = 327), 33.2% for 0–5 years vs 38.3% for 5–12 years, p = 0.0336). The prevalence of danger signs was significantly higher among mRDT positive children (40.6%) compared to children diagnosed with scabies (2.9%) (p<0.001). Notably, we observed a huge variation in the proportions of children with severe hypoxemia and IMCI danger signs between facilities. The proportions of

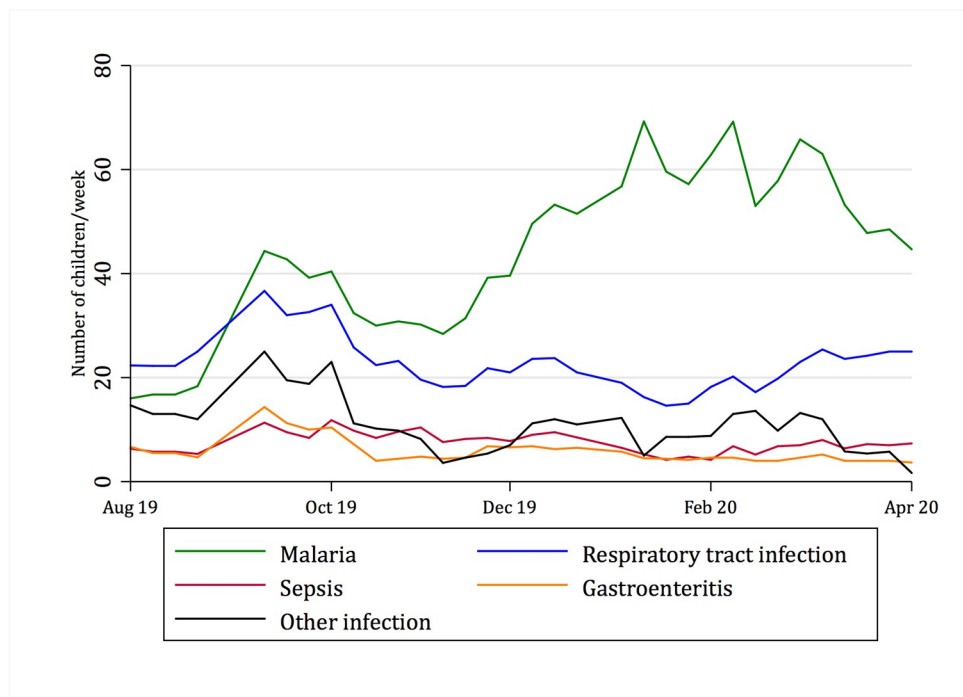

**Fig 1. Seasonality of different infections by week.**

children with danger signs ranged between facilities from 4.1–66.0% (median 18.8%). Severe hypoxemia was reported in 0–3.7% of patients (median 0.3%)–excluding the two research assistants with suspected invalid results (reporting 15.2% and 32.5% hypoxemia, respectively) and moderate hypoxemia 0–26.9% (median 2.5%) (S1 Fig).

## Temporal trends

Fig 1 demonstrates the seasonality of reported infectious diagnoses over the study period, and Fig 2 presents the seasonality of reported infectious versus non-infectious illness along with the prevalence of IMCI danger signs, low oxygen saturation and low blood glucose concentrations. While there is a clear seasonality in infectious disease presentations, with one peak almost exclusively caused by malaria, the prevalence of danger signs, hypoglycemia and hypoxemia are relatively stable throughout the year.

## Discussion

This study assessed the prevalence of low oxygen saturation and low blood glucose concentrations among children seeking outpatient care in Malawi. Our results demonstrate an overall low prevalence of WHO defined hypoglycemia and hypoxemia amongst a general pediatric outpatient population. Yet, moderate hypoglycemia and hypoxemia were more common (11.1% and 5.4%, respectively). More than a quarter of the included children presented with any IMCI danger sign. The presence of a danger sign identified all children with severe hypoglycemia, but only 23.5% of children with severe hypoxemia. The variance in proportions of hypoxemia and IMCI danger signs between facilities suggests issues in the accuracy of assessments.

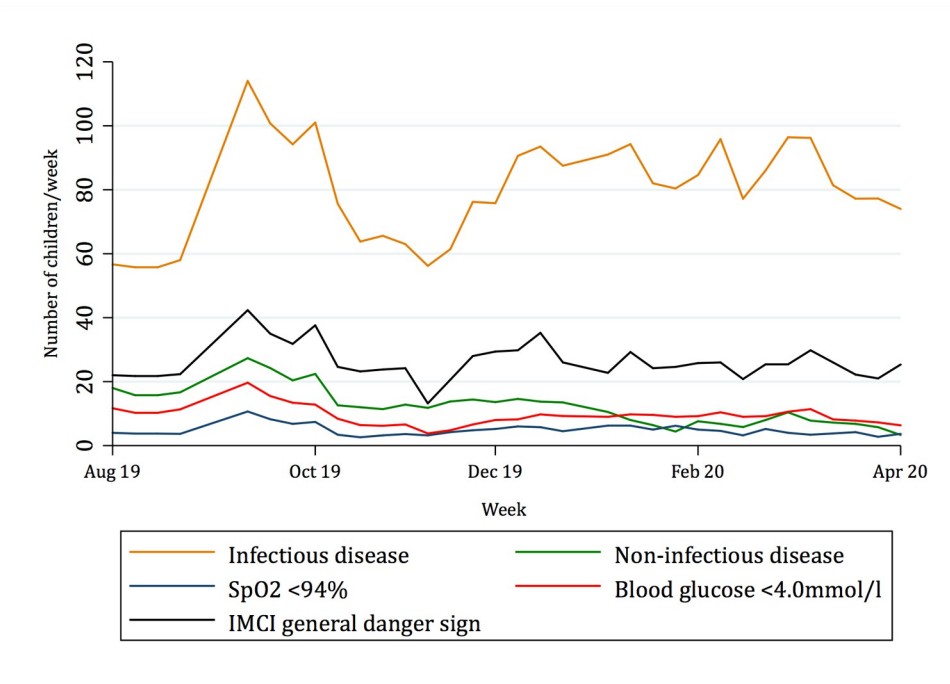

**Fig 2. Seasonality of infectious disease, non-infectious disease, IMCI danger signs, SpO$_2$ <94% and blood glucose <4.0mmol/l.**

There is a lack of data on the prevalence of hypoxemia in undifferentiated populations of sick children presenting to primary health care facilities in low-income countries, with earlier studies focusing on hypoxemia prevalence in children admitted to hospital, or in children diagnosed with pneumonia [23]. A minority of children in our study were diagnosed with pneumonia, despite acute respiratory tract infections being commonly diagnosed. Notably, none of the children diagnosed with pneumonia were severely hypoxemic and only 15.6% had moderate hypoxemia. When combined, the hypoxemia prevalence among pneumonia and non-pneumonia respiratory tract infection did not differ from the overall study population. Unsurprisingly, the hypoxemia prevalence was much lower in this outpatient population than in studies on hypoxemia in children in hospital settings, where prevalences of 5.3–14.1% have been reported [23–25].

WHO defined hypoglycemia (<2.5mmol/l) was rare in this population, with only four patients affected. Nonetheless, a substantially higher prevalence of moderate hypoglycemia (11.1%) was found. The WHO definition of hypoglycemia as a blood glucose concentration of <2.5mmol/l is a topic of debate, following a number of studies showing that sick children with blood glucose concentrations above 2.5mmol/l and with a variable upper limit of as high as 5.0mmol/l, also suffer an increased risk of mortality compared to children with higher blood glucose concentrations [11, 15, 16]. While there is an absence of comparable data from primary care facilities, severely sick children with moderate hypoglycemia are likely to benefit from a higher level of monitoring [26] and it can be hypothesized that referral of children with moderate hypoglycemia could contribute to reduced mortality.

All children with severe hypoglycemia presented with an IMCI danger sign, in line with a study from Laos in which 93.3% of admitted children with hypoglycemia also presented with any IMCI general danger sign [27]. The danger signs lethargy and inability to drink/feed were

more common among children with low blood glucose concentrations. However, the specificity of any general danger sign for detecting hypoglycemia was low since severe hypoglycemia affected only four children but a general danger sign was present in almost 29% (n = 848). Two thirds of the moderately hypoglycemic children had no danger sign, so consequently were not likely to be referred. Glucose testing at frontline facilities has the potential to identify the children with moderate hypoglycemia without danger signs, who may still benefit from referral for observation and/or treatment.

In contrast to hypoglycemia, our findings correspond with previous research showing that IMCI danger signs inadequately identify severely hypoxemic children in outpatient settings [14, 28].Chest indrawing has been suggested as the single best predictor of hypoxemia in children with pneumonia but still with only 69% sensitivity and 82% specificity [29]. In this real-world context, the presence of chest indrawing was not commonly reported by health workers and did not improve the identification of hypoxemic children. This likely reflects sub-optimal guidelines implementation and poor quality of respiratory clinical examination [30], but is reflective of routine care and subsequent case management. These data reinforce the need for pulse oximetry for the detection of hypoxemia, in a general population of acutely sick children [31]. A recently published data-linkage study found that pulse oximetry at outpatient facilities identified fatal pneumonia cases that would not have been referred using WHO referral guidelines only [32].

Considering the increased mortality risk among children with moderate hypoxemia and hypoglycemia it is possible that early identification and hospital referral of these children could prevent progression to severe hypoglycemia/hypoxemia and reduce mortality. However, the potential benefits need to be weighed against the risk of adding patients to already over-burdened hospitals and the costs for individual families implied by potential unnecessary referrals. Therefore, issues related to the quality of assessment are of high relevance for future guideline considerations. Despite training and close supervision, we chose to exclude results from two out of 20 research assistants due to unrealistically high proportions of hypoxemia. Known challenges in achieving reliable measurements with pulse oximetry include motion artifacts, poor perfusion, irregular heart rhythm, skin pigmentation, probe positioning and presence of abnormal hemoglobin molecules [33–36]. We also noted a higher proportion of unstable curves among younger children in this study. Considering that it may be more challenging to achieve a stable curve in a severely sick child, a failed $SpO_2$ measurement has been suggested as a referral criteria [14]; but it is also important to consider the potential for false positives leading to unnecessary referrals if quality cannot be maintained.

The prevalence of IMCI danger signs was similar between age groups, and no difference was seen between children under 5 and children aged 5–12 in the presence hypoxemia or hypoglycemia, suggesting that IMCI danger signs may be applied even in older children. The overall prevalence of danger signs was high and raised concerns for over-classifications by health workers. However, the significant difference in IMCI danger signs prevalence among children with scabies and those with a positive malaria RDT indicates that danger signs are assigned to "the right patients", which is also supported by previous findings [37]. There was a clear seasonality in cause of presentation, with a defined malaria peak during the rainy season (January–March), as would be expected. The peak of mixed infections seen during the southern hemisphere winter is also in line with expected respiratory pathogen transmission, although suggestive of misdiagnosis of pneumonia as malaria [38, 39]. However, the prevalence of danger signs, as well as hypoxemia and hypoglycemia were relatively stable throughout the year, and is surprising given the wide variation in patient loads observed in the hospital during the year.

Adherence to other aspects of the IMCI assessment however showed poor performance, such as the lack of respiratory rate recorded for children with reported cough and almost a third lacking recorded nutritional status. This is in line with previous studies from Sub-Saharan Africa, in which a correct classification/diagnosis was achieved by health workers' assessments in around 40% of the children [30, 40, 41].

We had four key limitations, first that this district wide study relied primarily on capturing data from standard assessments of sick children, which may result in inter-center variability due to differences in staff performance. We chose to use this approach to get a picture of routine rather than optimized guideline use, but data quality concerns were seen. Secondly, in the absence of validated danger signs for children aged 5–12 years we applied the same danger signs as for children under 5 years. Thirdly, the lack of outcome and follow-up data is a limitation since no conclusion can be drawn on the risks of different levels of oxygen saturation or glucose concentration. Finally, the research assistants were non-clinical staff and while they received training, supervision and follow-up, it was apparent that there were still issues in some of the oximetry assessments. This raises an important point when designing programs for oximetry scale-up in primary care, and the need for on-going mentorship.

## Conclusion

The prevalence of hypoxemia and hypoglycemia at primary care level are low, but moderate levels are not uncommon and could potentially be useful as an added tool to determine referral needs. IMCI danger signs were present in children with hypoglycemia, but not in the majority with severe or moderate hypoxemia. This highlights the difficulties in diagnosing hypoxemia through clinical presentation only, especially in a setting where lack of training, equipment and time is common. Future studies should focus on the role of moderate hypoxemia and hypoglycemia in subsequent outcomes, and optimized case management for these children.

## Supporting information

**S1 Table. IMCI general danger signs (unconscious, sleepy/lethargic, convulsing, vomits everything, unable to eat/drink) depending on clinical characteristics.** $SpO_2$ results excluded from manuscript included. * Respiratory rate was assessed in 399 children. ** 3 caregiver refused $SpO_2$-meassurment. *** No result due to refusal by caregiver (3), lack of test strips (22) and child too agitated (1).
(DOCX)

**S1 Fig. Prevalence of low $SpO_2$-results (<90%, 91–93%) and IMCI danger signs depending on health center (HC) and hospitals (H).** *The results of $SpO_2$ from HC 2 and HC 9 were excluded from the analysis.
(TIF)

## Acknowledgments

We would like to thank all the children and their parents who participated in this study, and the healthcare workers who supported our research assistant teams in their work. We are also grateful to the research assistants for their hard work, and the Mchinji District Health Management Team for their input and support.

## Author Contributions

**Conceptualization:** André Thunberg, Beatiwel Zadutsa, Carina King, Josephine Langton, Charles Makwenda, Helena Hildenwall.

**Data curation:** Beatiwel Zadutsa, Everlisto Phiri.

**Formal analysis:** André Thunberg.

**Funding acquisition:** Helena Hildenwall.

**Methodology:** Beatiwel Zadutsa, Carina King, Helena Hildenwall.

**Project administration:** Beatiwel Zadutsa, Lumbani Banda, Charles Makwenda.

**Supervision:** Beatiwel Zadutsa, Everlisto Phiri, Carina King, Lumbani Banda, Charles Makwenda, Helena Hildenwall.

**Writing – original draft:** André Thunberg, Carina King, Helena Hildenwall.

**Writing – review & editing:** Beatiwel Zadutsa, Everlisto Phiri, Josephine Langton, Lumbani Banda, Charles Makwenda.

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
