## [Decision Letter · Decision Letter 0]

18 Nov 2021

PGPH-D-21-00696

Hypoxemia, hypoglycemia and IMCI danger signs in pediatric outpatients in Malawi

Dear Dr. Thunberg,

Thank you for submitting your manuscript to PLOS Global Public Health. After careful consideration, we feel that it has merit but does not fully meet PLOS Global Public Health’s publication criteria as it currently stands. Therefore, we invite you to submit a revised version of the manuscript that addresses the points raised during the review process.

We look forward to receiving your revised manuscript.

Kind regards,

Claire E. von Mollendorf

Academic Editor

Journal Requirements:

1. Please provide additional details regarding participant consent. In the ethics statement in the Methods and online submission information, please describe how verbal consent was documented and witnessed, and why written consent was not obtained.

2. In the online submission form, you indicated that "Unidentified data used in this study is available upon request from the main authors"

3. In your financial disclosure statement,  state the initials, alongside each funding source, of each author to receive each grant.

Reviewers' comments:

Reviewer's Responses to Questions

**Comments to the Author**

1. Does this manuscript meet PLOS Global Public Health’s publication criteria? Is the manuscript technically sound, and do the data support the conclusions? The manuscript must describe methodologically and ethically rigorous research with conclusions that are appropriately drawn based on the data presented.

Reviewer #1: Partly

Reviewer #2: Yes

2. Has the statistical analysis been performed appropriately and rigorously?

Reviewer #1: No

Reviewer #2: No

3. Have the authors made all data underlying the findings in their manuscript fully available (please refer to the Data Availability Statement at the start of the manuscript PDF file)?

Reviewer #1: No

Reviewer #2: Yes

4. Is the manuscript presented in an intelligible fashion and written in standard English?

Reviewer #1: Yes

Reviewer #2: No

5. Review Comments to the Author

Reviewer #1: REVIEWER COMMENTS

Abstract

1. Line 26: include the word “of” after “prevalence”

2. Line 32: may be you can call them research assistants instead of data collectors

3. Line 33 and 34: include in brackets the manufacturers of the respective equipment that was used ie Lifebox LB-01 pulse oximeter and AccuCheck Aviva glucometers

4. Line 35 and 36: the author mentions severe and moderate hypoxemia and hypoglycaemia and then gives respective ranges. It is best to dissociate the moderate from severe hypoxemia or hypoglycaemia to make more clarity in the sentence

5. Conclusions: I am not sure whether the analysis is robust enough to draw these conclusions.

6. The authors say that they run a chi2 test… but the outcome(s) of the study are not clear.

Introduction: main text

7. Line 60 – 62: It would be better to link the success of the IMCI program to most recent targets like the SDGs instead of the long gone MDGs

8. The objective of the study is not quite clear.

9. It would have been great to classify the IMCI in terms of the illnesses

10. Materials and methods: it would be great to describe the Emergency paediatric treatment and referral in Malawi in frontline healthcare settings – EREMISS study)study in which the current research study is nested. If the paper for this study is published then give reference to the paper. When was the EREMISS study conducted? Was the EREMISS study conducted concurrently with the study under review? What kind of participants were recruited? What was the eligibility criteria? How long was the parent study?

11. Study setting: It is important to describe the different types of study sites involved in the study.. do the patients pay for the services provided? What is the structure of the healthcare setting in Malawi? How do the different study sites fit in with in the health system structure of Malawi? What kinds of patients seek health care at the different study sites in general? Do these patients pay for the services offered? What is the difference between “health centre” and “hospital”? this distinction can be described earlier on in the text so that the reader can understand the results table 1 much better.

12. Include a separate section on sample size estimation in which the authors can state that all children seen in a certain time period were included.

13. What was the rationale for only including only participants that had an acute illness? Why were those classified as having moderate, severe illness not recruited yet they could be having characteristics that may be of interest to science and policy

14. Participant recruitment: line 111 – you can refer to the data collectors as research assistants.

15. Line 116 – 118: It is not possible that it is the children that were approached for recruitment. Actually the parents or care givers of these children are the ones who should have been approached to obtain consent for the recruitment of their children. What do the ethical guidelines say in Malawi about the assenting of children? Some of these children should have been able to assent. These procedures should be discussed in detail.

16. Data collection procedures: line 124: please include the manufacturer of the Lifebox LB-01 pulse oximeter and the machine for blood glucose, the AccuCheck Aviva point of care device.

17. How was observer bias handled while the measurements of variables was being done? This can be clarified in data collection procedures.

18. What was the qualification of the “data collectors” (research assistants)? Were they trained in handling children? Were they nurses? Midwives? Or doctors? Were they trained in research? Were the research assistants employees of the different health facilities at which the study was conducted?

19. Was consent sought for the collection of blood from the child (it is a separate consent process from the participation in the study)

20. For those children that were hypoxemic and those that were hypoglycemic, what further steps did the study team under take to ensure that these participants received the medical care that they required? This should be highlighted for purposes of equipoise

21. Line 138: where was the data being uploaded? Was it on a server?

22. It would be very informative to include a section entitled variables under which the authors describe all the key / important variables that they have measured in the study ie outcome variables and exposures.

23. Line 169 – 170: the IMCI guidelines recommend the evaluation of a number of danger signs ie bacterial infection, dehydration, jaundice, diarrhoea, low blood sugar, difficulty in breathing etc. It is important for the authors to specify exactly which danger signs they included in their study and how this differed from the exposures that they objectively measured.

24. Ethical statement: line 179 – 183: this study involved paediatric participants. The authors must show an effort to convince the readership that the participation of these participants was not coerced. If the data collectors were employees of the different study sites and at the same time research assistants, how did this double role play prevent coercion of participation? How was confidentiality of the information collected from the participants observed? Was the information collected by the ODK uploaded to a server? If so, how was this uploaded data handled? Was assenting for children aged 10 years or more done? What if this child declined assenting? How did the authors handle care givers or parents of children / infants that were minors (less than 18 years)? In this section the authors can also endeavour to make mention of what was done for oh how the study handled participants that needed medical attention. In the results the authors say that some parents / care givers declined participation. How were these participants handled?

25. Results: was data about the father of the child collected? It would be informative to include this in the analysis as well. What reference was used to classify the disease diagnosis? Was it the ICD? The authors can mention this.

26. The authors should calculate or estimate the 95%CI for the major proportions / prevalence. This can be done in stata using the exact method.

27. The results can be presented in a more coherent manner. This section can be improved after the section of the variables is constructed.

28. Discussion: this is too long and can be shortened.

29. The study must be having some limitations. The authors can look into including this section in the manuscript. For example this is a purely descriptive study and it’s very difficult to make inferences to the general population.

30. THE AUTHORS SHOULD CONSIDER TO USE THE STROBE GUIDELINES WHILE REVISING THIS MANUSCRIPT

Reviewer #2: GENERAL

This is an interesting and valuable topic.

The data is original but there are numerous errors in the tables and the analysis is incomplete.

IMCI was developed specifically for the assessment of children below five years of age but in this study the danger signs have been applied to children up to 12 years of age without any justification or comment on the appropriateness or applicability of the danger signs to the older child. The analysis of danger signs and the relationship of these to both hypoxemia and hypoglycaemia needs to be disaggregated by age to differentiate between those under and over five years of age.

The English is fair although there is the occasional typographical error.

TITLE

The title is an appropriate reflection of the content of the paper.

ABSTRACT

The abstract is a fair summary of the full content of the article.

However it may need to be revised if further analysis reveals any differences between younger and older children.

There are a few errors in the text:

Line 32 - needs to be corrected to “one day per month in each facility”

Line 34 – WHO needs to be presented in full before the abbreviation can be used

Line 42 – need to add severely ie “23.5% of the severely hypoxemic and..”

INTRODUCTION

The introduction is clear and succinct.

It sets the scene, highlights common challenges in identifying and referring critically ill children from the primary care level and suggests a possible response to identifying these children.

This study was undertaken to assess the validity of this suggestion.

METHODOLOGY

Although the description of the methodology is clear it requires some expansion or additions:

1. Justification is required for the use of IMCI danger signs out of their intended purpose ie in the assessment of children over 5 years of age. They need to provide some evidence to show that this is an appropriate tool to use in older children.

2. Some explanation is required as to why the authors chose to supplement the 5 IMCI danger signs with temperature, respiratory rate and chest indrawing and what evidence supports the choice of these specific signs as “danger” signs

Line 106 – “hospitals” is the incorrect term as participants were enrolled from hospitals and health centres. Better to refer to “health facilities”.

RESULTS

The results are presented in a series of 4 tables.

Tables 1, 2, and 3 all contain data errors and only table 4 has any statistical analysis.

Comments on individual tables are very limited and are restricted to the prevalence of the focus area of each table (hypoxemia, hypoglycaemia or danger sings) without any commentary on possible associations with or variations between subgroups ie age, gender, diagnosis or nutritional stuats.

Table 1 – Baseline characteristics

A note should be added to indicated that children may have more than 1 diagnosis – there were 2,943 children in the study but 3,379 diagnoses.

There are 2 data errors - The number of mothers with secondary education who presented at a health centre should be 400 not 340 and the total number of children with a diagnosis of other infection should be 268 not 358 and the % needs to be adjusted from 12.2% to 9.1%

A superficial analysis suggests that children who presented to the hospital were younger, had mothers with more education, a known nutritional status and more sepsis but fewer respiratory infections. Some statistical analysis is required to determine whether any of these are significant.

Table 2 - Hypoglycemia

This table has 5 errors all in the final column for total numbers – I’m assuming this is the error as the numbers in the other columns do not match the totals.

Total number of well nourished children should be 1669 not 1671

Total number of children with missing nutritional status should be 925 not 949

Total number of children with a temperature ≥37.5oC should be 890 not 892

Total number of children with gastroenteritis is 187 not 188

Total number of children with a diagnosis of “other infections” is the correct sum of previous columns but is inconsistent with the number in Table 1 which was 268

Again there is no commentary on patterns with subgroups – children with moderate hypoglycaemia appear to be older, female, and to be sleepy and unable to drink

Table 3 - Hypoxemia

This table has 7 errors all in the final column for total numbers – again I’m assuming this is the error as the numbers in the other columns do not match the totals.

Total number of well nourished children should be 1472 not 1671;

Total number of children with moderate malnutrition should be 130 not 154

Total number of children with severe malnutrition should be 140 not 169

Total number of children with missing nutritional status should be 893 not 949

Total number of children with a temperature ≥37.5oC should be 818 not 892

Total number of children with gastroenteritis is 155 not 188

Total number of children with a diagnosis of “other infections” is 270 not 358 and this is also inconsistent with the number in Table 1 which was 268

Again there is no commentary on patterns with subgroups – children with moderate hypoxemia appear to be younger males with convulsions or loss of consciousness

Table 4 – Danger signs

There are no obvious errors in this table however a bit more information is needed – there were 848 children with danger sings which is 28.8% but how many had more than one danger sign, what were the number of children with danger signs in each age group.

Whilst interesting I don’t believe that the temporal trends and Figures 1a and 1b add any value.

Figure S1 needs a title and correction of the legend – IMCI refers to danger sings so relabel either as “IMCI danger signs” or just “”danger signs” rather than “IMCI”

DISCUSSION

The discussion is reasonable however the sequence should ideally mirror the order in which data is presented in the results section ie hypoglycaemia then hypoxemia and then IMCI danger signs.

The local findings are compared to a few studies of either in- or outpatient settings from LMIC however there is no comment on the age groups in these studies compared to the age group in this study. Presumably the outpatient studies using IMCI focus on the under-5 age group which is not the case in this study. So some comment is required on the age groups and how this may have affected the findings.

There are a few “typos”

Lines 307, 309 and 323 refer to severe hypoglycaemia

Line 323 sign not sig

Line 344 in a general population

LIMITATIONS

This is adequately covered apart from the issue of using an assessment tool designed for children under 5 years of age on children aged 5 – 12 years.

CONCLUSIONS

The conclusion is appropriate.

REFERENCES

There is an extensive, comprehensive and up to date list of 44 references.

There are however inconsistencies in the structure of the references with some giving the first page number only and others the first and last pages.

6. PLOS authors have the option to publish the peer review history of their article (what does this mean?). If published, this will include your full peer review and any attached files.

**Do you want your identity to be public for this peer review?** For information about this choice, including consent withdrawal, please see our Privacy Policy.

Reviewer #1: No

Reviewer #2: No

---

## [Decision Letter · Decision Letter 1]

25 Jan 2022

PGPH-D-21-00696R1

Hypoxemia, hypoglycemia and IMCI danger signs in pediatric outpatients in Malawi

Dear Dr. Thunberg,

Thank you for submitting your manuscript to PLOS Global Public Health. After careful consideration, we feel that it has merit but does not fully meet PLOS Global Public Health’s publication criteria as it currently stands. Therefore, we invite you to submit a revised version of the manuscript that addresses the points raised during the review process.

We look forward to receiving your revised manuscript.

Kind regards,

Claire E. von Mollendorf

Academic Editor

Journal Requirements:

Additional Editor Comments (if provided):

There are a few spelling and grammatical errors in the manuscript. Please proofread prior to resubmission.

Consider adding a comment in the manuscript regarding the missing nutritional status data and the differences between the healthcare center and hospital sites.

Reviewers' comments:

Reviewer's Responses to Questions

**Comments to the Author**

1. If the authors have adequately addressed your comments raised in a previous round of review and you feel that this manuscript is now acceptable for publication, you may indicate that here to bypass the “Comments to the Author” section, enter your conflict of interest statement in the “Confidential to Editor” section, and submit your "Accept" recommendation.

Reviewer #2: (No Response)

2. Does this manuscript meet PLOS Global Public Health’s publication criteria? Is the manuscript technically sound, and do the data support the conclusions? The manuscript must describe methodologically and ethically rigorous research with conclusions that are appropriately drawn based on the data presented.

Reviewer #2: Yes

3. Has the statistical analysis been performed appropriately and rigorously?

Reviewer #2: Yes

4. Have the authors made all data underlying the findings in their manuscript fully available (please refer to the Data Availability Statement at the start of the manuscript PDF file)?

Reviewer #2: Yes

5. Is the manuscript presented in an intelligible fashion and written in standard English?

Reviewer #2: Yes

6. Review Comments to the Author

Reviewer #2: GENERAL COMMENT

The data errors have been corrected and the inconsistencies explained.

Comment has been added on the use of IMCI danger signs in the older, 5 – 12 year, age group and a comparison of findings between young and older age groups is now included.

I believe that the principle of presenting words in full before using acronyms or abbreviations holds and applies to WHO which needs to be spelt out in full at the start of the document.

The English is fair.

METHODOLOGY

The authors have now provided a rationale for the use of IMCI danger signs in older children.

However they still need to provide an explanation as to why they chose to supplement the 5 IMCI danger signs with temperature, respiratory rate and chest indrawing. They have provided this in their response to the reviewers but did not include this in the article. It needs to be included in the article.

RESULTS

The data has been cleaned and inconsistencies explained.

Line 246 close the brackets (49.2%)

Line 321 “health worker” should be “research assistants”

DISCUSSION

This is reasonable and now includes comment on the different age groups.

LIMITATIONS

This has been expanded to include a comment on the use of an assessment tool designed for children under 5 years of age on children aged 5 – 12 years.

7. PLOS authors have the option to publish the peer review history of their article (what does this mean?). If published, this will include your full peer review and any attached files.

**Do you want your identity to be public for this peer review?** For information about this choice, including consent withdrawal, please see our Privacy Policy.

Reviewer #2: No

---

## [Editor Report · Decision Letter 2]

2 Mar 2022

Hypoxemia, hypoglycemia and IMCI danger signs in pediatric outpatients in Malawi

PGPH-D-21-00696R2

Dear Thunberg,

We are pleased to inform you that your manuscript 'Hypoxemia, hypoglycemia and IMCI danger signs in pediatric outpatients in Malawi' has been provisionally accepted for publication in PLOS Global Public Health.

Best regards,

Claire E. von Mollendorf

Academic Editor